# Site-Directed Mutagenesis Mediated by Molecular Modeling and Docking and Its Effect on the Protein–Protein Interactions of the bHLH Transcription Factors SPATULA, HECATE1, and INDEHISCENT

**DOI:** 10.3390/plants14121756

**Published:** 2025-06-08

**Authors:** Pablo López-Gómez, Daniela De La Mora-Franco, Humberto Herrera-Ubaldo, Corina Díaz-Quezada, Luis G. Brieba, Stefan de Folter

**Affiliations:** 1Advanced Genomics Unit (UGA-Langebio), Center for Research and Advanced Studies (Cinvestav), Irapuato 36824, Mexico; pablo.lopez@cinvestav.mx (P.L.-G.); daniela.delamora@cinvestav.mx (D.D.L.M.-F.); hh585@cam.ac.uk (H.H.-U.); corina.diaz@cinvestav.mx (C.D.-Q.); luis.brieba@cinvestav.mx (L.G.B.); 2Rosario Izapa Experimental Field of National Institute of Forestry, Agriculture and Livestock Research (INIFAP), Chiapas 30870, Mexico; 3Department of Plant Sciences, University of Cambridge, Cambridge CB2 3EA, UK

**Keywords:** AlphaFold2, HawckDock, protein structure modeling, flower, gynoecium, transcription factor, protein interactions, SPATULA, HECATE1, INDEHISCENT, site-directed mutagenesis

## Abstract

The aim of this study was to investigate the biological relevance of predicted sites involved in protein–protein interaction formation by bHLH transcription factors associated with gynoecium development in Arabidopsis (*Arabidopsis thaliana*). We used AlphaFold2 to generate three-dimensional protein structures of the bHLH proteins SPATULA (SPT), HECATE1 (HEC1), and INDEHISCENT (IND). These structures were subjected to molecular docking using the HawkDock server, enabling the identification of potential interaction sites. PCR-based site-directed mutagenesis was used to modify the predicted interaction sites, followed by testing for protein–protein interaction formation using Bimolecular Fluorescence Complementation (BiFC) assays. Furthermore, these modified versions were overexpressed in Arabidopsis to observe whether gynoecium and fruit development would be affected. BiFC assays with the modified versions revealed a complete loss of the SPT-HEC1 interaction and a strong reduction in the SPT-IND interaction. The overexpression experiments in Arabidopsis showed that the *35S::SPT-4A* line exhibited strong phenotypes in the development of the medial tissues of the gynoecium, resulting in reduced seed number and shorter fruits. In the *35S::HEC1-2A* line, a reduced seed number and shorter fruits were also observed, but no other obvious defects were observed. Finally, the *35S::IND-3A* line was less affected than the *35S::IND* line. In the latter, medial tissue development was strongly affected, while in the *35S::IND-3A* line, it was only slightly affected; however, a reduced seed number and shorter fruits were observed. In summary, the predicted interaction sites are relevant and, when modified, affect gynoecium development in Arabidopsis. The findings demonstrate that predictive computational tools represent a viable strategy for a deeper understanding of protein–protein interactions.

## 1. Introduction

One of the most complex structures produced by Arabidopsis (*Arabidopsis thaliana*) is the gynoecium [1,2]. For its correct development, it is necessary that different tissues acquire specific identities in a certain time and space. The function of this structure or organ is to protect the ovules, favor pollination and fertilization of the ovules, and disperse the seeds at the end of fruit development [1,3]. The involvement of transcription factors is key for flower and fruit development [4]. Furthermore, an important aspect of transcription factors is the formation of protein–protein interactions (PPIs), which enables them to fulfill their roles [5,6,7].

PPIs are defined as the physical contact between two proteins that occurs selectively in a particular biological context [8,9]. These types of interactions are responsible for a wide range of processes, including cell–cell interactions, control of metabolic processes, and, especially developmental processes, as part of a regulatory network. A single protein hardly acts as a solitary entity when performing its function. It is estimated that about 80% of proteins act in complexes [10,11]. One of the questions in this field of study is the biological relevance of these interactions. One of the approaches that could help us to answer this question is to have structural information. For this, it is necessary to have the three-dimensional structure of the proteins of interest.

Obtaining the three-dimensional structure of a protein represents a challenge. The relevance lies in the fact that from the three-dimensional structure of a protein, it is possible to infer aspects related to its biological and molecular functions. Currently, around 200,000 protein structures have been described using techniques such as nuclear magnetic resonance (NMR), X-ray crystallography, and cryogenic electron microscopy (cryo-EM) [12,13]. However, there is still a large gap between the number of available protein sequences and the number of protein structures obtained. To overcome the need for information about the structure of a protein, predictive or sequence modeling techniques, in combination with computational techniques, are an alternative [14,15]. One of the major advances in recent years is AlphaFold, a program development that uses a neural network to perform structural predictions [16]. In 2021, AlphaFold2 was released, which uses a computational method based on machine learning that incorporates physical and biological information of proteins to predict three-dimensional protein structures [17].

The availability of these tools opens up the possibility of scaling up studies to include molecular docking of proteins. Molecular docking refers to a computational approach used to predict how two molecules, usually a small molecule (a drug or ligand) and a macromolecule (such as a protein or nucleic acid), interact with each other to form a stable complex [18,19]. Regarding the prediction of PPIs, several computational tools have been developed, whose methods combine knowledge from the fields of biology, chemistry, and quantum physics on protein sequences, structures, and their folding. Some tools and platforms available online are MDockPP [20], LZerD [21], Multi-LZerD [22], Cluspro [23,24], HADDOCK [25], and HawkDock [26]. The latter generates models of protein complexes based on the decomposition of their binding free energies (MM/GSBA), the ATTRACT docking algorithm, and the HawkRank score. With the above parameters, this platform generates at least ten models of the most favored complexes in terms of energy, and in the best model it is possible to determine the residues that could be involved in the interaction, both in the receptor and in the ligand [26].

With the availability of all these new tools, we wondered whether we could predict sites involved in the PPI formation of proteins involved in gynoecium development in Arabidopsis. In this work, we report on the use of AlphaFold2 for the molecular modeling of protein structures [17] in combination with the molecular docking tool HawkDock [26], allowing the prediction of possible interaction sites involved in PPI formation between two proteins. Subsequently, to functionally analyze whether the predicted sites are involved in PPI formation, we used PCR-based site-directed mutagenesis, followed by bimolecular fluorescence complementation (BiFC) assays, to evaluate whether PPI formation was altered. Finally, we generated Arabidopsis overexpression lines with the site-directed mutagenized gene versions to observe whether gynoecium and fruit development were affected. For this study we selected three bHLH transcription factors involved in gynoecium development: namely, SPATULA (SPT), necessary for the correct development of tissues such as the carpel margins, septum, transmitting tract, style, and stigma [27,28,29]; HECATE1 (HEC1), required for the correct development of the septum, transmitting tract, and stigma [30]; and INDEHISCENT (IND), which is necessary for the specification of carpel and valve margin tissues [31,32]. It has been reported that to fulfill these functions in gynoecium development in Arabidopsis, SPT interacts with HEC1 and IND, as detected by yeast two-hybrid (Y2H) and BiFC techniques [30,32,33,34,35].

## 2. Results

### 2.1. Protein Structure Modeling and Molecular Docking

Our research interest is gynoecium development and the involvement of transcription factors. Furthermore, the formation of protein–protein interactions (PPIs) between these transcription factors is important (e.g., ref. [35]). In this work, we focused on three bHLH transcription factors that are important for the development of medial tissues in the gynoecium, namely SPT, HEC1, and IND [3,28,30]. To obtain insights into the importance of specific contact sites for PPI formation between these proteins, we first used the AlphaFold2 platform to predict 3D protein structures for SPT, HEC1, and IND [17,36]. The modeling of SPT clearly showed the bHLH domain composed of residues 197–246 (Appendix A; in blue), in addition to a helix–loop–helix structure that comprises amino acids 45–59 (Appendix A; in red). On the other hand, the rest of the structure did not show good confidence scores (Appendix A). In the case of the HEC1 and IND protein structures, the known bHLH domains could be clearly detected (Appendix A). Furthermore, as seen for the predicted SPT structure, besides the bHLH domain and a helix–loop–helix structure, the rest of the structure prediction is considered flexible or poorly modeled (Appendix A).

Next, the modeled structures were used for molecular docking using the HawkDock platform [26]. In all cases, the first model of each 3D structure prediction by AlphaFold2 was used for molecular docking and was prepared with the addition of charges using Chimera 1.18 [37]. As a result of the molecular docking between SPT-HEC1 and SPT-IND with HawkDock, the first 10 generated models were analyzed, and the best model was selected based on the decomposition of their binding free energies (MM/GSBA), the ATTRACT docking algorithm, and the HawkRank score from HawkDock [26] (Appendix A). To determine which sites to mutate to disrupt the SPT-HEC1 PPI interaction, model 4 was selected, as it had the lowest total dimer binding free energy of −46.17 Kcal mol^−1^ (Appendix A), while for SPT-IND, model 2 was selected, with the lowest total dimer binding free energy of −43.58 Kcal mol-1 (Appendix A). In the analysis of the interaction sites between SPT-HEC1 (Figure 1a), the possible binding sites involved in SPT were found to be amino acids Val176, Val177, Asp178, Ile372, and Glu171, while in HEC1 they included the amino acids Arg138, Arg143, His139, Arg123, and Lys130 (Table 1).

For the SPT-IND dimer (Figure 1b), the possible amino acids involved in SPT were Glu179, Asp178, Phe130, Ser133, and Ala180, whereas for IND, they included the amino acids Arg113, Arg139, Ser135, Arg131, and Lys142 (Table 2).

Based on these results, using site-directed mutagenesis, we choose to substitute the amino acids with the lowest free binding energy that were next to each other. Therefore, for SPT, we decided to mutate the triplets that code for the four residues Val176-Glu179; for HEC1, the two sites Arg138 and His139; and for IND, the three amino acids Asn112-Arg114. We designated the following names for the mutated versions: SPT-4A, HEC1-2A, and IND-3A. All the changes were made towards alanine (Ala) using a PCR-based method [38]. The amino acid Ala is non-polar, with a small side chain, minimizing the impact on the structure of the protein, making alanine substitutions a good option to investigate the importance of specific residues in protein–protein interactions.

### 2.2. Protein–Protein Interaction Analyses Using the BiFC Assay

As previously described, SPT interacts with HEC1 [30,33,34] and IND [32]. In order to determine the effect of site-specific mutations in triplets coding for amino acids predicted to be involved in dimer formation, PPI formation was analyzed using Bimolecular Fluorescence Complementation (BiFC) assays, carried out using the Arabidopsis mesophyll protoplast system [39,40,41]. In the case of SPT-HEC1 dimer formation (Figure 2a), it was observed that the interaction was negatively affected in each combination with a mutated version (*VN:SPT-4A VC:HEC1-2A*, *VN:SPT-4A VC:HEC1*, *VN:SPT VC:HEC1-2A*). In all combinations, no YFP signal was observed, in contrast to the combination with the unmodified protein versions (*VN:SPT VC:HEC1*), which showed an average interaction frequency of 34.22%, i.e., protoplasts with YFP signal (Figure 2b). In summary, PPI formation is completely lost when *SPT* or *HEC1* is mutated.

In the case of SPT-IND dimer formation, PPI formation was affected when mutated versions were used, as reduced frequencies of protoplasts with YFP signal were observed (Figure 3). The observed frequency of protoplasts with YFP signal when the unmodified protein versions were used (*VN:SPT VC:IND*) was 27.83%. In contrast, for the combinations with mutated protein versions *VN:SPT-4A VN:IND-3A*, *VN:SPT-4A VC:IND*, and *VN:SPT VC:IND-3A*, average interaction frequencies of 14.91%, 13.51%, and 12.99%, respectively, were observed, being significant differences with respect to the control. In summary, PPI formation was reduced roughly by 50%.

### 2.3. Generation of Overexpressing Lines of Site-Directed Mutagenized Genes

The BiFC assays demonstrated that PPI formation was lost or reduced when site-specific mutated versions of SPT, HEC1, or IND were used. The next question to answer was whether these site-directed mutagenized genes would generate morphological effects when overexpressed in Arabidopsis plants. For this, the non-mutated and mutated gene versions were expressed under the constitutive 35S promoter in Arabidopsis plants (Appendix A). Increased gene expression in all lines compared to wild-type Arabidopsis plants was confirmed (Appendix A).

### 2.4. Functional Characterization of the 35S::SPT-4A Line

We analyzed first the effect of overexpression of the site-directed mutant version of the *SPT* coding sequence during flower and fruit development in Arabidopsis. Regarding flower appearance and structure, there were no notable differences between the wild type, *35S::SPT* and *35S::SPT-4A* lines (Figure 4a). On the other hand, when analyzing fruit development, clear phenotypes were observed in the *35::SPT-4A* line. It was found that *35S::SPT-4A* plants produced smaller fruits (8.22 ± 1.73 mm) than wild-type plants (14.78 ± 0.64 mm), and plants of the *35S::SPT* line (15.64 ± 1.60 mm) (Figure 4b,e). Another aspect that was affected in the *35S::SPT-4A* line was the average number of seeds per fruit (9.88 ± 5.84 seeds/fruit); highly significant differences were observed in the average number of seeds per fruit when comparing the wild-type plants (48.68 ± 3.46 seeds/fruit) and the *35S::SPT* line (49.53 ± 11.11 seeds/fruit) (Figure 4b,f). Interestingly, the shape of the gynoecium (apical fusion defect; Appendix A) and silique (spatula-shaped), and the reduction in seed number of the *35S::SPT-4A* line are similar to what has been reported for the *spt-2* loss-of-function mutant [28]. As expected, both the average fruit size and the average number of seeds per fruit produced by the *35S::SPT* line were not different from those produced by the wild type, as has been reported before [42,43].

To better understand the phenotypes observed in terms of fruit development and seed production in the *35S::SPT-4A* line, histological sections of the inflorescences were carried out and compared against the wild type and the *35S::SPT* line. The sections were stained with neutral red and alcian blue staining to observe transmitting tract differentiation. In gynoecia of the *35S::SPT-4A* line, no fusion of the septum was observed, compared to what was observed in the sections of the wild type and the *35S::SPT* line (Figure 4c and Appendix A). Furthermore, based on the neutral red and alcian blue staining, no transmitting tract differentiation was observed in the *35S::SPT-4A* line (Figure 4c and Appendix A). In addition, staining of pollen tubes with aniline blue was carried out in hand-pollinated gynoecia. As expected, gynoecia from the *35S::SPT-4A* line had no transmitting tract, and therefore pollen tube growth was strongly reduced. This effect was seen both with pollen from the *35S::SPT-4A* line and with wild-type pollen (Figure 4d). In addition, these phenotypes have been reported for the *spt-2* loss-of-function mutant [28].

### 2.5. Functional Characterization of the 35S::HEC1-2A Line

One of the other well-studied interactors of SPT is HEC1 [30,33]. In general, no obvious morphological changes in terms of flower structure in the *35S::HEC1* or the *35S::HEC1-2A* line were evident with respect to the wild type (Figure 5a). As for fruit development, significant differences were observed among the three lines. The plants of the *35S::HEC1-2A* line produced the smallest fruits (11.25 ± 1.62 mm), while the plants of the *35S::HEC1* line had fruits with an average size of 13.84 ± 0.89 mm, which is significantly different from the average fruit size of the wild-type plants, measured at 14.78 ± 0.64 mm (Figure 5b,e). In terms of seed yield per fruit, significant differences were also observed among the three lines. Plants of the *35S::HEC1-2A* line produced the lowest number of seeds per fruit with 24.40 ± 9.42, followed by plants of the *35S::HEC1* line, with an average of 44.03 ± 7.01 seeds per fruit, compared to the wild type, which had an average of 48.68 ± 3.46 seeds per fruit (Figure 5b,f).

As performed for the *SPT* overexpressing lines, histological sections were prepared and stained with neutral red and alcian blue. No defects were observed during gynoecium development (Figure 5c and Appendix A). In the same line, septum development and transmitting tract differentiation were observed in wild-type plants and in both *HEC1* lines (Figure 5c). As with pollen tube growth, no clear differences were observed in either the *35S::HEC1* or the *35S::HEC1-2A* lines, compared to the wild type (Figure 5d).

### 2.6. Functional Characterization of the 35S::IND-3A Line

Another well-studied interactor of SPT is IND [32]. The mutated *IND* gene was overexpressed in the Arabidopsis Col-0 background and contrasted with the wild type and the *35S::IND* line. Overall, flower appearance exhibited no difference in the *35S::IND-3A* line compared to the flowers produced by wild-type plants. However, the flowers of the *35S::IND* line showed flower alterations, such as the presence of carpelloid sepals and loss of floral organ abscission (Figure 6a). Similar phenotypes were observed in a previously reported *35S::IND* line [32] and also in mutants with defects in polar auxin transport, such as *pin-formed* (*pin1*) and *pinoid* (*pid*) [44]. Regarding fruit development, differences were observed in the average fruit length of the *35S::IND* (9.81 ± 2.17 mm) and *35S::IND-3A* (8.73 ± 1.68 mm) lines when compared to the wild type (14.78 ± 0.64 mm), while no differences were observed in fruit length between the two overexpressing lines (Figure 6b,e).

As for seed production per fruit, both the *35S::IND* and the *35S::IND-3A* lines were affected, with an average of 16.01 ± 8.98 and 17.86 ± 6.10 seeds per fruit, respectively, compared to those produced by wild-type plants, with an average of 48.68 ± 3.46 seeds per fruit (Figure 6b,f).

To better understand these phenotypes, histological sections of gynoecia from the three lines were prepared again and stained with neutral red and alcian blue. In the *35S::IND-3A* line, no evident defects were observed at early stages, but at stage 12, the alcian blue staining was slightly reduced, indicating less development of the transmitting tract (Figure 6c and Appendix A). On the other hand, though not the main focus of this study, in the *35S::IND* line at stage 12, septum fusion appeared to be affected; moreover, no transmitting tract was visible (Figure 7c). Pollen tube growth was also analyzed by aniline blue staining, and a reduction was observed in both overexpression lines compared to the wild type. In the case of the *35S::IND-3A* line, the pollen tubes advanced only 40% into the ovary after hand-pollination, and when hand-pollinated with wild-type pollen, the pollen tubes advanced around 50% into the ovary. Similarly, the pollen tubes only advanced around 40% into the ovary of the *35S::IND* line (Figure 7c).

### 2.7. Morphological Analysis of the Style Width in the Overexpression Lines

The style is an important tissue for plant reproduction. Initial observations suggested the presence of some fine differences in the style width. Therefore, detailed measurements of the style width were performed at all lines. We observed a reduction in the width of the style of gynoecia from the *35S::SPT-4A* line (221.8 ± 11.71 µm), which showed a significant difference with the style width of gynoecia from the *35S::SPT* line and the wild-type, with style widths of 290.3 ± 14.30 and 279 ± 11.41 µm, respectively (Figure 7a,b). When analyzing style width in the *HEC1* lines, we found a difference only in the style width of gynoecia produced by the *35S::HEC1-2A* line, with an average of 255 ± 12 µm, compared to the wild type (279 ± 11.41 µm) (Figure 6c,d). Style width was also analyzed in the *IND* lines. Style width was not affected in gynoecia from the *35S::IND-3A* line (288.7 ± 8.71.mm) compared to the wild-type (279 ± 11.41 mm). However, the style width of gynoecia from the *35S::IND* line was affected, with an average of 240.4 ± 14.51 mm (Appendix A).

## 3. Discussion

### 3.1. Molecular Modeling and Docking for the Prediction of Protein–Protein Interaction Sites 

In this work, we evaluated the models generated by AlphaFold2 in combination with HawkDock-mediated molecular docking, with the aim to perform site-directed mutagenesis on predicted sites involved in the PPIs of three bHLH transcription factors, which are important for tissue development in the gynoecium of Arabidopsis. Regarding the molecular modeling, in the three structures modeled (SPT, HEC1, and IND), the highest confidence in modeling was observed in the predicted basic helix–loop–helix (bHLH) domain (Appendix A). The regions without correct modeling can be considered as disordered regions. For instance, for the SPT sequence, disordered regions have been previously described in UniProt [45], predicted using the MobiDB-lite method [46]. In the cases of the modeled structures of HEC1 and IND (Appendix A), the bHLH domains predicted by the PFscape [47], and the PROSITE protocol, respectively [48], could be clearly detected, in addition to a disordered region at the N-terminal end of IND, as described by the automatic annotation using the sequence analysis-SAM method [45].

Several methods have been described for the prediction of protein–protein interaction sites. The first methods were based on multiple sequence alignments; however, with this approach, it is difficult to generate good results, since a family of transcription factors usually consists of related proteins resulting from duplication events [49]. However, searches for specific and repetitive motifs have allowed progress in this area [50,51].

Here, the use of HawkDock-mediated molecular docking with modeled structures of complete protein sequences gave rise to good predictions of sites involved in PPI formation. In contrast, in our hands, using the HDOCK server [52], which uses only the conserved domains, resulted in fewer predicted PPI sites.

### 3.2. The Mutated Sites Are Important for the Formation of SPT-HEC1 and SPT-IND Dimers

In this work, we found that the mutated versions of the SPT-HEC1 dimer completely lost their ability to interact, and when the mutated versions of the SPT-IND dimer were used, PPI formation was reduced to around 50%. These results strongly suggest that the predicted sites are involved in PPI formation.

For the SPT protein, the triplets that code for Val176, Val177, Asp178, and Glu179 were all substituted for alanine (Ala), which is a neutral, non-polar amino acid. This mutated region is located one residue downstream of the conserved acidic domain (Glu-161-Glu-174), which appears to be important for the function of SPT in gynoecium development [43]. The four mutated amino acids are also partially conserved [43]. Another characteristic, the mutated SPT region is flanked by Ala-175 and Ala-180, and since alanine substitutions were made, the mutated SPT version contains now a region of six continuous alanines (Ala175-Ala180). This could have drastic effects on the PPI capacity of SPT, since replacing these sites with alanine implies eliminating the side chains [38]. Indeed, the PPI formation of the mutated SPT version is clearly altered.

In the case of HEC1, the substituted sites to Ala were Arg138 and His139, which are located at the N-terminal end of the bHLH domain. The exact position of the mutations is at residues 11 and 12 of the basic region, which has been considered relevant for nuclear localization (NLS) in other members of the bHLH family, such as SPT and ALCATRAZ (ALC) [43,53]. Therefore, it will be interesting to evaluate whether this region is involved in the nuclear localization of HEC1 in future experiments.

In the IND mutation, the substituted sites to Ala were Asn112, Arg113, and Arg114. This region is located just four residues upstream of the bHLH domain, and it belongs to a disordered region, according to the Uniprot automatic sequence analysis annotation (SAM) [45]. As observed, the interaction detected between IND and SPT was reduced by around 50%. This may suggest that more sites are involved in PPI formation, and, indeed, based on the molecular docking results, more sites were predicted (Table 2). Therefore, generating additional substitutions in IND-3A, such as the predicted site Arg139 (which has the lowest energy after Arg113; Table 2), will be interesting in the future.

### 3.3. Morphological Effects Due to Overexpression of Mutated Gene Versions

The *35S::SPT-4A* line appears to act as a dominant negative mutant, since the plants produced gynoecia and fruits with phenotypes similar to the loss-of-function *spt-12* mutant [42]. A possibility is that the SPT-4A protein version is still able to bind the DNA of its target genes but no dimer formation occurs; therefore, its regulatory action could be absent. The endogenous SPT version is present in the wild-type background; however, it apparently cannot function normally because of the observed phenotypes. Dimer formation is important for proteins belonging to the bHLH family [54]. For this purpose, it would be interesting to carry out experiments to see if or how SPT-4A binding to SPT targets is affected, or, for instance, if it is able to form non-functional homodimers with the endogenous SPT protein. On the other hand, as has been reported, overexpression of non-modified SPT does not affect the development of the gynoecium or fruit of Arabidopsis [42,43]. This further highlights the importance of understanding how overexpression of the mutated version of SPT generates such a contrasting phenotype.

In the case of the *35S::HEC1-2A* line, the mutated version displayed a phenotype with reduced fertility compared to the wild-type, although this does not appear to be related to septum or transmitting tract development. Pollen tube growth was unaffected, as also observed in the single *hec1* mutant [30]. In this case, the presence of the redundant paralogous HEC2 and HEC3 could be helping to counteract the effect of HEC1-2A. Therefore, it would be interesting to introduce the *HEC1-2A* version into the *hec1,2,3* triple mutant [30].

In the *35S::IND-3A* line, no strong effects were observed. The IND-3A version showed reduced PPI formation with SPT. This might cause a reduction in target gene binding. Despite the endogenous IND present, maybe not much IND protein or IND dimer is needed to carry out its function; therefore, not many effects were seen when overexpressed. A better insight into the effect of IND might be obtained by introducing the modified version in the *ind* mutant [32]. Actually, most phenotypes were observed when overexpressing the wild-type *IND* version, as has been reported [32]. It is also important to consider that the overexpression of *IND* favors the expression of *SPT* because *SPT* is a target gene of IND [32]. Perhaps the IND-3A version is not able to affect the expression of *SPT*. Future research can provide insight into the molecular mechanisms.

Future experiments, which could be considered for a deeper understanding of the results obtained in this work, would be generating double overexpression lines (SPT-4A-HEC1-2A and SPT-4A-IND-3A). Furthermore, the effect of these mutations could also be studied by in situ BiFC studies, by expressing the mutated versions under their native promoters [55]. Protein stability could also be explored using, e.g., antibodies. On the other hand, to better understand the effects on the regulatory action of the different mutated versions of single proteins or the SPT-4A-IND-3A dimer versions, their DNA binding affinity could be analyzed by electrophoretic mobility assays (EMSA) [56]. On the other hand, structural insight might also be obtained from modelling and docking experiments using the mutated gene sequences. For instance, using Alphafold2, the structure of SPT-4A showed slight changes, whereas the structures of HEC1-2A and IND-3A remained unchanged (Appendix A). Furthermore, Alphafold3 is now available, offering more developed algorithms and enabling both modeling and docking within the same tool [57].

Finally, this work shows the potential of protein structural modeling and molecular docking for an in-depth study of PPIs. At present, few studies have been reported using this strategy to study flower and fruit development. One example involves MADS-box transcription factors, where directed mutagenesis of the SEPALLATA (SEP) members revealed that the formation of SEP-mediated tetramers was affected, leading to poor flower development [58]. However, instead of predictions, crystallographic structures of the proteins were used. Another recent study used a strategy that was more similar to the work presented here (modeling and docking), in which the mutations were directed to leucines, predicted to be involved in the interaction between SEP3 and APETALA1 (AP1). The overexpression of the modified versions in Arabidopsis, resulted in prolonged vegetative growth, increased leaf size and leaf number, and modifications in floral structures [59]. In conclusion, the approach used here can provide insights into the discovery of sites involved in PPI formation and their possible relevance in plant development.

## 4. Materials and Methods

### 4.1. Structure Modeling and Molecular Docking

The three-dimensional structures of SPT (AT4G36930), HEC1 (AT5G67060), and IND (AT4G00120) proteins were modeled in AlphaFold2 [17]. Protein sequences were obtained from UniProt and modeled using AlphaFold2 via ColabFold [37,60] (https://colab.research.google.com/github/sokrypton/ColabFold/blob/main/AlphaFold2.ipynb?authuser=1 (accessed on 28 June 2022)), generating five relaxed structures per protein using the PDB100 template mode. Default MSA settings were applied, while advanced settings were adjusted to set the maximum interactions to 0; figures were rendered at 300 dpi; all other parameters remained at default values [37,60]. The first model for each was selected, and charges were added to the structures using Chimera 1.18 [37], followed by molecular docking on the HawkDock server platform [26]. Docking was performed for the interactions between SPT-HEC1 and SPT-IND, with the SPT structure being always considered as receptor and the protein structures of HEC1 and IND as ligands. The obtained protein complexes were visualized in PyMOL 3.1.3 [61], and the distances between amino acids with greater probability of contact based on free energy of binding were determined.

### 4.2. Site-Directed Mutagenesis

The coding sequence of the *SPT*, *HEC1,* and *IND* genes was amplified and cloned into the entry vector pENTR™-D-TOPO (Invitrogen by ThermoFisher Scientific, Carlsbad, CA, USA).

The site-directed mutant of SPT was generated substituting amino acids Val176, Val177, Asp178, and Glu197 with 4× Ala. To generate the site-directed mutant for HEC1, the amino acids Arg138 and His139 were substituted by 2× Ala. To generate the site-directed mutant for IND, the amino acids Asn112, Arg113 and Arg114 were substituted by 3× Ala. The primers used are listed in Appendix A.

These mutations were obtained using the PCR-based Q5 site-directed mutagenesis protocol with some modifications (New England Biolabs, Ipswich, MA, USA). Briefly, the mutants were obtained by PCR with Q5 polymerase, each in a reaction of 12.5 μL, and the pENTR clone of each gene was used as template. Subsequently, each PCR product was phosphorylated by making a 15 μL reaction containing 3 μL of the PCR product and 1 μL of the enzyme PKN (ThermoFisher Scientific TM, Vilnius, Lithuania); the reaction was incubated at 30 °C for 20 min. Phosphorylated modified genes were ligated with 0.5 μL of T4 DNA ligase (ThermoFisher Scientific TM, Vilnius, Lithuania) and incubated at 16 °C for 16 h. To each ligation, 1 μL of DpnI (New England Biolabs, Ipswich, MA, USA) was added and incubated at 37 °C for 2 h. From the previous reaction, 4 μL was used to transform *E. coli* DH5α. Plasmid purification was carried out, and positive clones were verified by sequencing. From here on, the positive clones of the mutated versions of *SPT*, *HEC1,* and *IND* will be named *SPT-4A* (V176A-V177A-D178A-E179A), *HEC1-2A* (R138A-H139A), and *IND-3A* (N112A-R113A-R114A), respectively.

### 4.3. BiFC Assay

The pENTR clones of *SPT*, *SPT-4A*, *HEC1*, *HEC1-2A*, *IND*, and *IND-3A* were recombined with the Gateway-compatible destination vectors pVN/gw and pVC/gw [62]. These vectors contain the N-terminal fragment composed of residues 1 to 154 (YN) and the C-terminal fragment composed of residues 155 to 238 (YC) of the Venus YFP variant, respectively. The backbone of these vectors is the transient expression plasmid pBI221 containing the 35S promoter [62]. The resulting expression vectors *VN:SPT*, *VN:SPT-4A VC:HEC1*, *VC:HEC1-2A*, *VC:IND*, and *VC:IND-3A* were isolated using the Zippy^®^ Plasmid Midiprep Kit (Zymo Research, Irvine, CA, USA). Fully expanded leaves from six-week-old *A. thaliana* Col-0 ecotype were collected and used for protoplast isolation following the “Tape-Arabidopsis-Sandwich” method [40,41]. The digestion step was carried out for 2–3 h. For expression vector transfection, 150 µL of protoplast solution containing 3 × 10^4^ cells was mixed with 10 µL of vector (1 µg µL^−1^ plasmid DNA). The interactions evaluated were *VN:SPT-VC:HEC1* as a positive control, *VN:SPT-4A-VC*-*HEC1-2A*, *VN:SPT-4A-VC:HEC1*, *VN:SPT-VC:HEC1-2A;* and *VN:SPT-VC:IND* as a positive control, *VN:SPT-4A-VC-IND-3A*, *VN:SPT-4A-VC:IND*, and *VN:SPT-VC:IND-3A;* as negative controls *VN:SPT*- and empty *VC:GW* were transfected (Appendix A). Transfection was performed following the PEG-Ca method [39,40,41]. The reaction was stopped after 15 min of incubation at room temperature. The analysis of Fluorescence restoration was carried out 16 h after transfection. Excitation wavelengths and emission filters were 488 nm/bandpass 505–530 nm for YFP, and 488 nm/band-pass 650–710 nm for chloroplast auto-fluorescence in a LSM800 confocal microscope (Zeiss, Wetzlar, DE, Germany). Four images per interaction were captured using a ×10 objective. For each image, the number of YFP-signal-positive protoplasts and the total protoplast count (approximately 200 per sample) were quantified. The frequency was calculated as: (YFP-positive protoplasts/total protoplasts) × 100.

### 4.4. Generation of Arabidopsis Overexpression Lines

The *35S::SPT* line was already available [42]. For the *35S::SPT-4A*, *35S::HEC1*, *35S::HEC1-2A*, *35S::IND*, and *35S::IND-3A* lines, the entry vectors with each gene were recombined with the binary overexpression pGD625 vector [63]. Recombination reactions were used to transform *E. coli* DH5α; subsequently, colonies were verified by PCR. Subsequently, these plasmids were used to transform *A. tumefaciens* GV3101. The colonies were verified by PCR and those that were positive were used to transform *A. thaliana* ecotype Col-0, using the floral dip method [64]. The transformed plants were selected by kanamycin resistance (50 μg mL^−1^). A summary of the obtained transgenic plants and those evaluated can be seen in Appendix A. Phenotyping was performed in the T2 generation; plants were grown in a greenhouse under standard growth conditions. Note: to observe the apical fusion defects in the gynoecium in the *spt* mutant, growth conditions were as follows: plants were grown in soil at ~23 °C under long-day conditions (16 h light/8 h darkness) in a growth chamber. The light source consisted of PHILIPS F17T8/TL841 fluorescent lamps, which emit warm white light with a color temperature of 4100K (neutral white), a high color rendering index (CRI ≥ 85), and ~50–100 µmol/m^2^/s of photon flux.

### 4.5. RNA Extraction and RT-PCR Analysis

RNA was extracted from flower buds of inflorescences of *35S::SPT*, *35S::SPT-4A*, *35S::HEC1*, *35S::HEC1-2A*, *35S::IND*, *35S::IND-3A*, and wild-type (Col-0) plants, as previously reported by [65], using the TRIzol reagent (Thermo Fisher Scientific, Aukland, New Zealand). The transgenic lines were in the T3 generation. Four biological replicates were sampled. A total of 1 μg of RNA, after DNAse I treatment, was used to synthesize cDNA using M-MLV reverse transcriptase according to the manufacturer’s protocol (Invitrogen by ThermoFisher Scientific, Carlsbad, CA, USA). The RT-PCR analysis (25 cycles) was performed using the primers listed in Appendix A with a MiniAmp™ Plus thermocycler (Applied Biosystems, Marsiling, Singapore). Quantification of PCR product intensity from gel was performed with ImageJ 1.8 (https://imagej.net/ij/ (accessed on 9 January 2025)).

### 4.6. Statistical Analysis

For the statistical analyses of fruit length and seed number, 6 fruits from ten homozygous plants of each line (*n* = 60) were collected. The fruits were collected from the main inflorescence, starting from fruit number 6 onward for each plant. Fruit length measurements were performed with ImageJ, and from the same fruits, the number of seeds was counted. The data were analyzed using a Kruskall–Wallis test, followed by a Dunn’s test. For style width, 10 gynoecia were analyzed, and the data were processed via an analysis of variance (ANOVA), followed by Tukey HSD (Honestly Significant Difference). All data analyses were performed in RStudio version 4.3.0.

### 4.7. Histology and Microscopy Analyses

For transverse gynoecia section analyses, the previously described method was followed [66]. Briefly, inflorescences of wild-type plants and all lines were collected and fixed in FAE solution (3.7% formaldehyde (*v*/*v*), 5% glacial acetic acid, and 50% ethanol) with vacuum (20 min, 4 °C) and incubated for 2 h at room temperature. The material was rinsed with 70% ethanol and incubated overnight at 4 °C in 70% ethanol, followed by dehydration in a series of ethanol dilutions (70%, 85%, 95%, and 100% ethanol) for 1 h each. Inflorescences were embedded in Technovit 7100 (Heraeus Kulzer, Hanau, Germany) according to the manufacturer’s instructions. Sections (12 to 15 μm thick) were obtained on a rotary microtome (Leica, Nussloch, Germany). To observe the transmitting tract, tissue sections were stained with a solution of 0.5% Alcian Blue and counterstained with 0.5% Neutral Red, as previously described [67]. A coverslip was placed over the sample, and pictures were taken using a bright-field microscope, Axio Imager.Z2 (Carl Zeiss, Jena, Germany; upright stand).

Pollen tube growth was analyzed using Aniline Blue staining, as described by [68]. Both wild-type and overexpressed lines were emasculated at the anthesis stage, and 24 h later hand-pollinated. The next day, the gynoecia were collected and fixed in absolute ethanol/acetic acid (3:1) for 2 h at room temperature. The fixed gynoecia were washed 3 times with distilled water and treated in softening solution of 8 M NaOH overnight. Then, the gynoecia were washed 3 times in distilled water and stained in Aniline Blue solution (0.1% Aniline Blue (*w*/*v*) in 150 mM K_2_HPO_4_ buffer, pH 11) for 4 h in the dark. Gynoecia were observed and photographed with a DM6000B fluorescence microscope under UV light (Leica).

## 5. Conclusions

The predictive tools in terms of molecular modeling mediated by AlphaFold2 and molecular docking generated by HawkDock represent a good complementary approach for studying the biological relevance of protein–protein interactions (PPIs). The predicted interaction sites for the SPT-HEC1 dimer are of biological significance, since PPI formation was completely lost. Overexpressing the mutated version of *SPT-4A* generated a loss-of-function-like phenotype, highlighting the biological relevance of the interaction of SPT with HEC1. The overexpression of the mutated version of *HEC1-2A* showed a partial impairment of fertility, which also highlights the relevance of this interaction. As for the formation of the SPT-IND dimer, a partial impairment of the interaction was observed in the BiFC assay (a reduction of roughly 50%), which might explain the weaker phenotype observed in the *35S::IND-3A* line compared to the *35S::IND* line.

## Figures and Tables

**Figure 1 plants-14-01756-f001:**
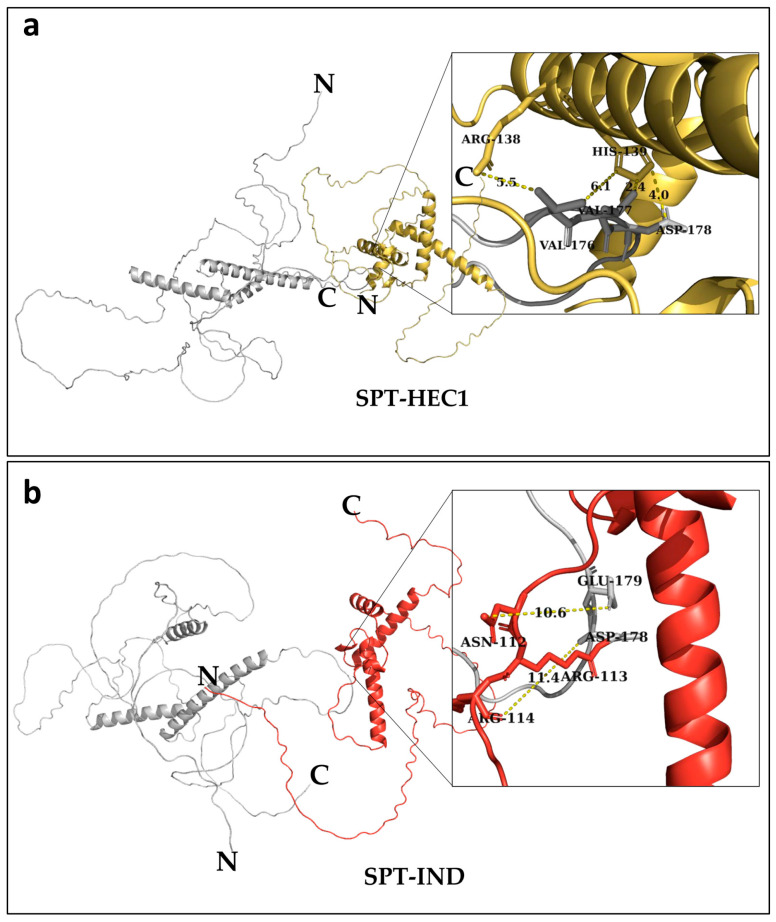
Representation of protein complexes of SPT obtained by molecular docking with HawkDock. (**a**) Protein complex SPT (gray)-HEC1 (yellow); in the zoomed area, contact sites are depicted; (**b**) protein complex SPT (gray)-IND (red); in the zoomed area, contact sites are depicted, and the numbers between the dotted lines indicate the distance in Ångström (Å). N- and C-termini are labeled.

**Figure 2 plants-14-01756-f002:**
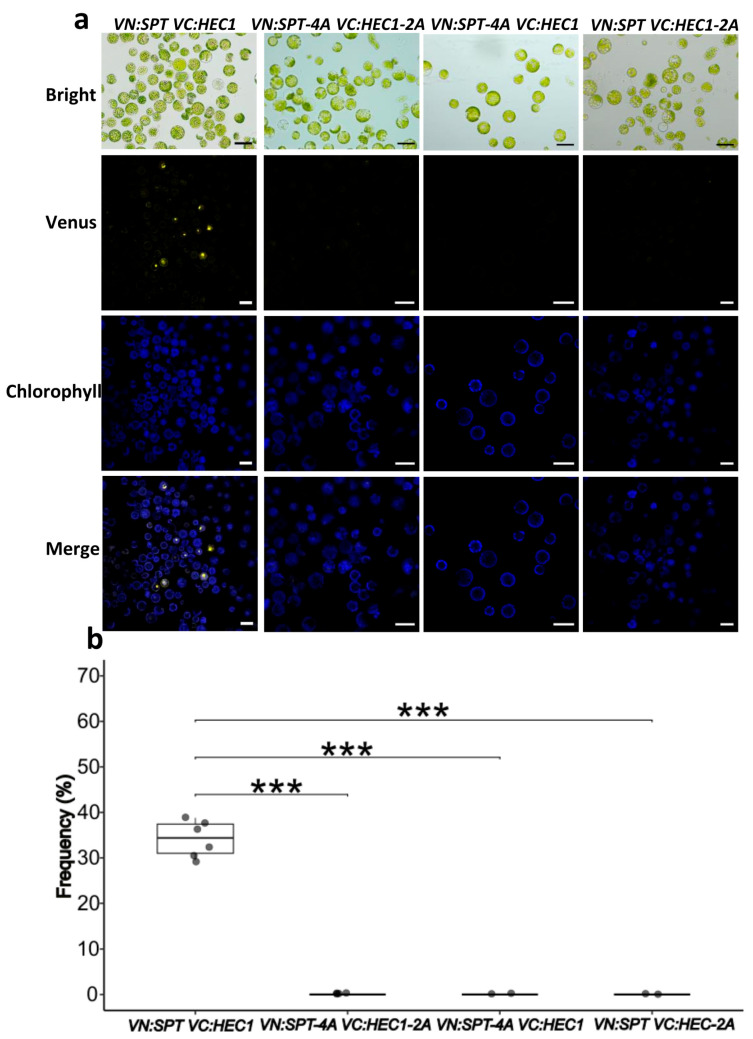
Analysis of PPI formation using the BiFC assay. (**a**) BiFC results of the SPT-HEC1 interaction and the mutant versions observed with a ×20 objective. (**b**) Quantification of BiFC results of the SPT-HEC1 interactions and its mutants. ANOVA, followed by Tukey’s test; *** *p* < 0.001, ns ≥ 0.05, *n* = 4 (one replicate ≈ 200 protoplasts in ×10 objective). Scale bars = 50 µm.

**Figure 3 plants-14-01756-f003:**
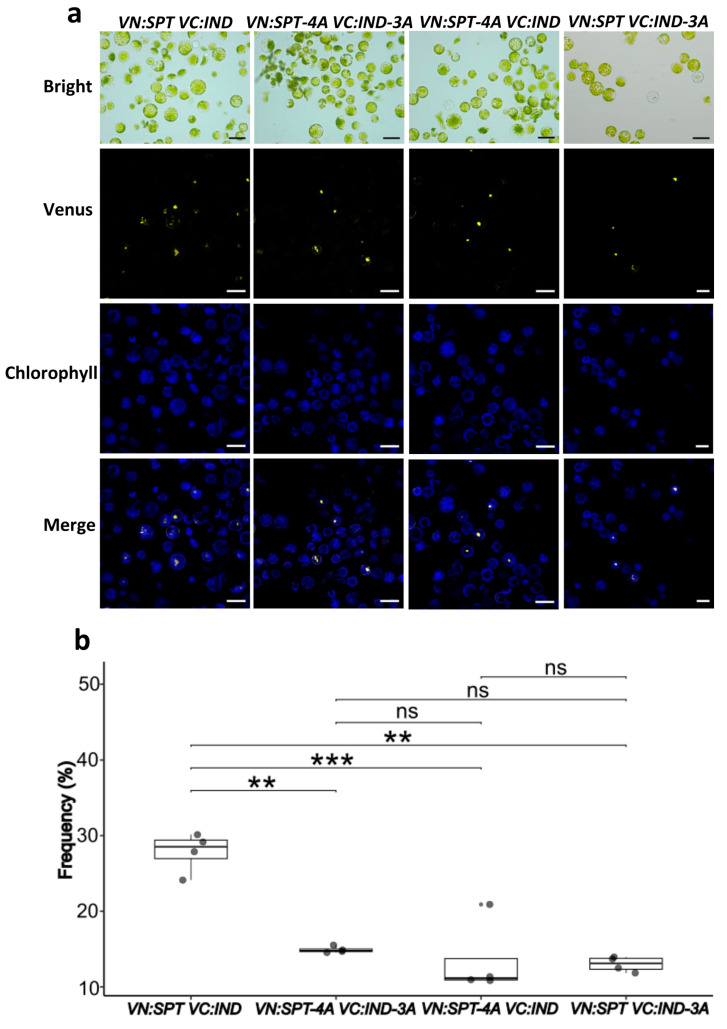
Analysis of PPI formation using the BiFC assay. (**a**) BiFC results of the SPT-IND interactions and the mutant versions observed with a ×20 objective. (**b**) Quantification of BiFC results of the SPT-IND interactions and its mutants. ANOVA, followed by Tukey’s test; *** *p* < 0.001, ** *p* < 0.01, ns ≥ 0.05, *n* = 4 (one replicate ≈ 200 protoplasts in ×10 objective). Scale bars = 50 µm.

**Figure 4 plants-14-01756-f004:**
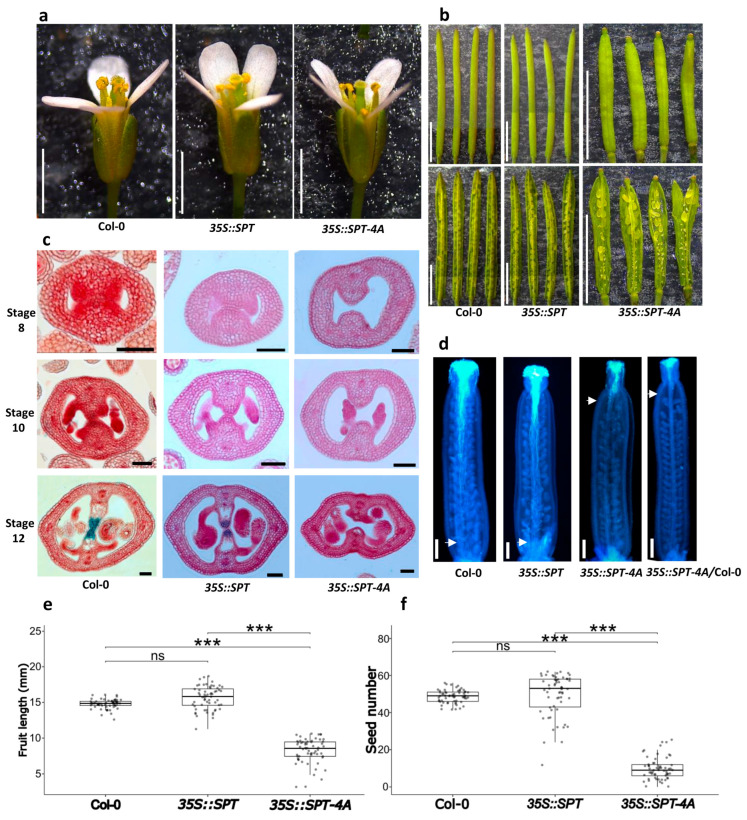
Functional analysis of the *35S::SPT-4A* line. (**a**) Flowers of wild-type plants (Col-0), *35S::SPT,* and *35S::SPT-4A*. (**b**) Fruits of wild-type plants (Col-0), *35S::SPT,* and *35S::SPT-4A*. (**c**) Gynoecia cross-sections stained with alcian blue and neutral red of wild-type plants (Col-0), *35S::SPT,* and *35S::SPT-4A* lines at stages 8, 10, and 12. Images were digitally extracted for comparison. (**d**) Pollen tubes stained with aniline blue of hand-pollinated gynoecia of wild-type plants (Col-0), *35S::SPT*, *35S::SPT-4A*, and *35S::SPT-4A* pollinated with Col-0 pollen. The white arrows mark the point where the majority of pollen tubes ceased growth. (**e**,**f**) Box plots showing analyses of fruit length (**e**) and seed number (**f**). Kruskal–Wallis test followed by Dunn’s test, *** *p <* 0.001, ns ≥ 0.05, *n* = 60. Note: images/measurements of wild-type (Col-0) samples are the same as in Figure 5 and Figure 6, except (**d**). Scale bars = 2 mm (**a**); 5 mm (**b**); 50 μm (**c**); 200 µm (**d**).

**Figure 5 plants-14-01756-f005:**
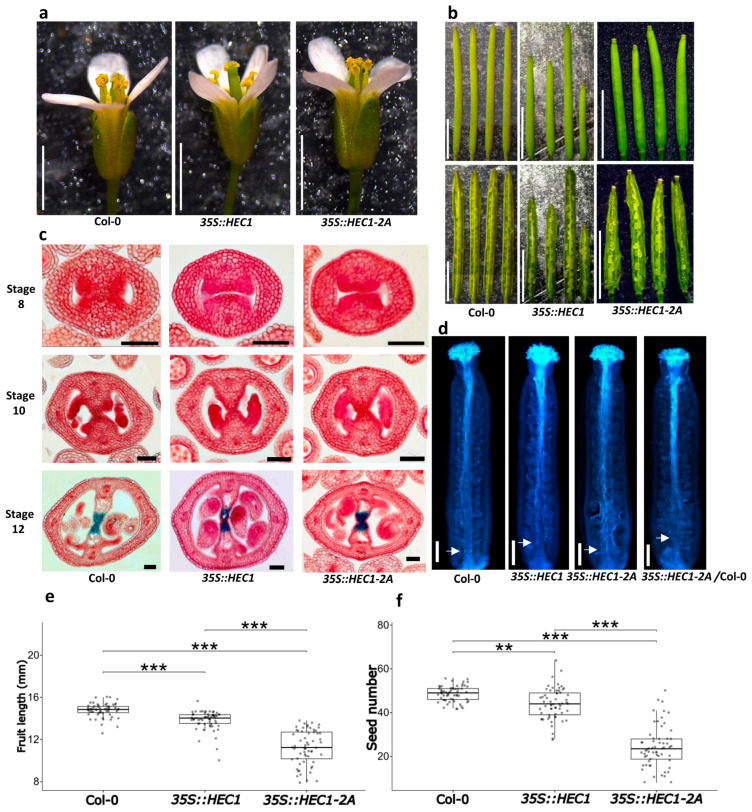
Functional analysis of the *35S::HEC1-2A* line. (**a**) Flowers of wild-type plants (Col-0), *35S::HEC1,* and *35S::HEC1-2A*. (**b**) Fruits of wild-type plants (Col-0), *35S::HEC1,* and *35S::HEC1-2A*. (**c**) Gynoecia cross-sections stained with alcian blue and neutral red staining of wild-type plants (Col-0), *35S::HEC1,* and *35S::HEC1-2A* lines at stage 8, 10, and 12. Images were digitally extracted for comparison. (**d**) Pollen tubes stained with aniline blue of hand-pollinated gynoecia in wild-type plants (Col-0), *35S::HEC1,* and *35S::HEC1-2A* pollinated with Col-0 pollen. The white arrows mark the point where the majority of pollen tubes stopped growth. (**e**,**f**) Box plots showing analyses of fruit length (**e**) and seed number (**f**). Note: images/measurements of wild-type (Col-0) samples are the same as in Figure 4 and Figure 6, except (**d**). Kruskal–Wallis test, followed by Dunn’s test; *** *p* < 0.001. ** *p* < 0.01, *n* = 60. Scale bars = 2 mm (**a**); 5 mm (**b**); 50 μm (**c**); 200 µm (**d**).

**Figure 6 plants-14-01756-f006:**
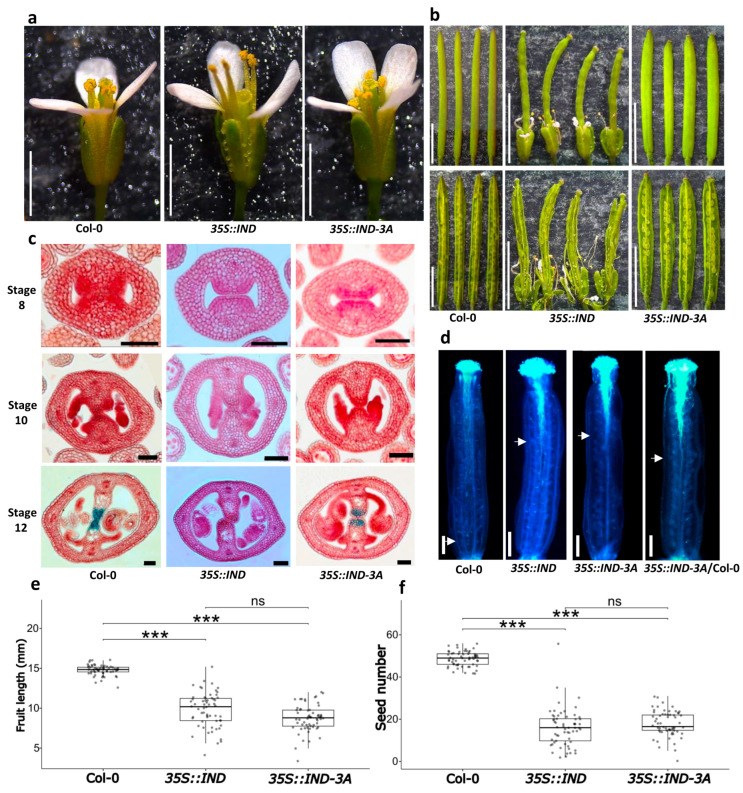
Functional analysis of the *35S::IND-3A* line. (**a**) Flowers of wild-type plants (Col-0), *35S::IND,* and *35S::IND-3A*. (**b**) Fruits of wild-type plants (Col-0), *35S::IND* and *35S::IND-3A*. (**c**) Gynoecia cross-sections stained with alcian blue and neutral red staining of wild type (Col-0), *35S::IND* and *35S::IND-3A* lines at stage 8, 10 and 12. Images were digitally extracted for comparison. (**d**) Pollen tubes stained with aniline blue of hand-pollinated gynoecia in the wild type (Col-0), *35S::IND* and *35S::IND-3A.* The white arrows mark the point where the majority of pollen tubes stopped growth. (**e**,**f**) Box plots showing analyses of fruit length (**e**), and seed number (**f**). Kruskal–Wallis test followed by Dunn’s test, *** *p* < 0.001, ns ≥ 0.05, *n* = 60. Note: images/measurements of wild-type (Col-0) samples are the same as in Figure 4 and Figure 5, except (**d**). Scale bars = 2 mm (**a**); 5 mm (**b**); 50 μm (**c**); 200 µm (**d**).

**Figure 7 plants-14-01756-f007:**
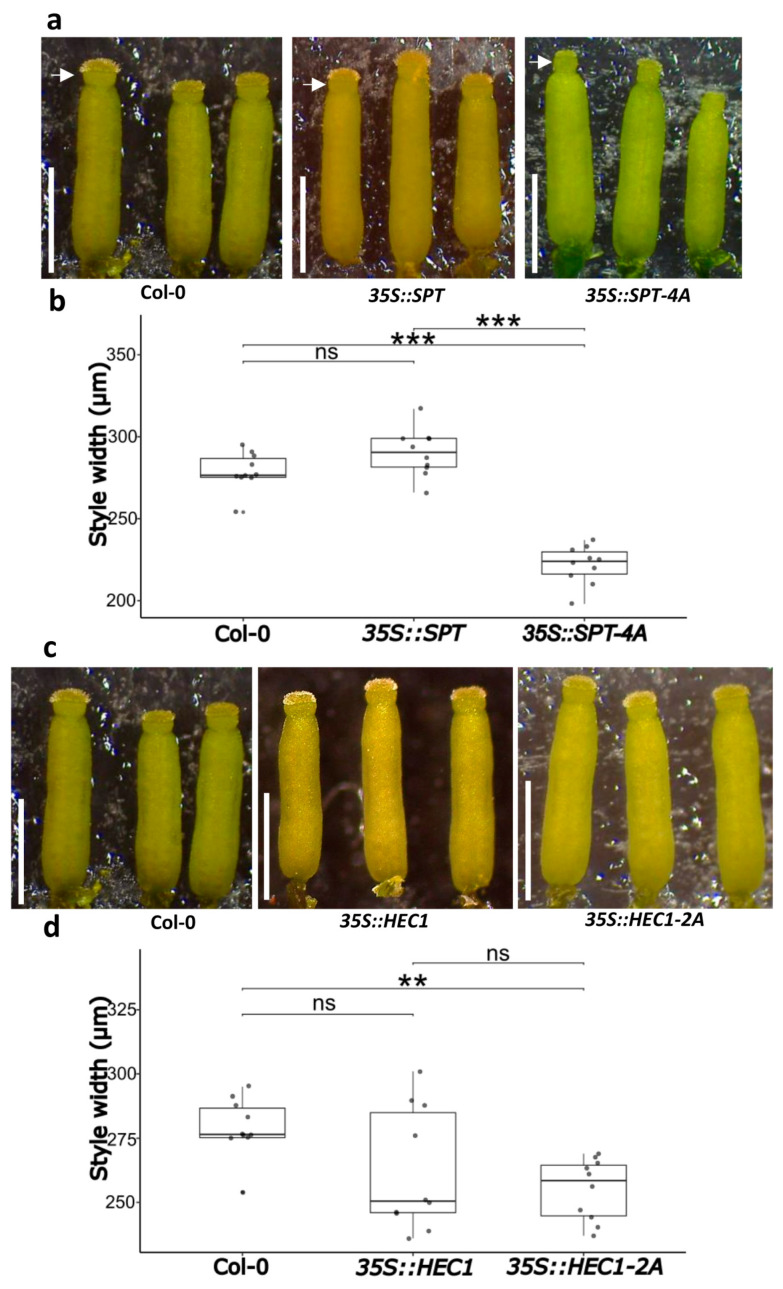
Functional analyses of *35S::SPT-4A* and *35S::HEC1-2A* gynoecia. (**a**) Gynoecia of wild-type plants (Col-0), *35S::SPT,* and *35S::SPT-4A*. White arrows denote style width measurement sites. (**b**) Quantitative analyses of style width in the gynoecia of wild-type plants (Col-0), *35S::SPT,* and *35S::SPT-4A.* (**c**) Gynoecia of wild type (Col-0), *35S::HEC1,* and *35S::HEC1-2A*. (**d**) Quantitative analyses of style width in the gynoecia of the wild type (Col-0), *35S::HEC1,* and *35S::HEC1-2A.* ANOVA, followed by Tukey’s test, **** p <* 0.001, *** p <* 0.01, ns ≥ 0.05, *n* = 10. Note: images/measurements of wild-type (Col-0) samples are the same as in Appendix A. Scale bars = 1 mm (**a**,**c**).

**Table 1 plants-14-01756-t001:** Top five most favored amino acids for dimer formation SPT-HEC1.

SPT	Binding Free Energy (Kcal mol^−1^)	HEC1	Binding Free Energy (Kcal mol^−1^)
Val177	−5.56	Arg138	−5.12
Val176	−4.29	Arg143	−4.75
Asp178	−3.44	His139	−3.04
Ile372	−2.06	Arg123	−3.0
Glu171	−1.95	Lys130	−2.91

**Table 2 plants-14-01756-t002:** Top five most favored amino acids for dimer formation SPT-IND.

SPT	Binding Free Energy (Kcal mol^−1^)	IND	Binding Free Energy (Kcal mol^−1^)
Glu179	−3.79	Arg113	−9.52
Asp178	−3.27	Arg139	−8.25
Phe130	−2.71	Ser135	−5.14
Ser133	−1.42	Arg131	−2.58
Ala180	−1.11	Lys142	−2.53

## Data Availability

The original contributions presented in this study are included in the article/Appendix A. Further inquiries can be directed to the corresponding author.

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
