# Peer review of "Site-Directed Mutagenesis Mediated by Molecular Modeling and Docking and Its Effect on the Protein–Protein Interactions of the bHLH Transcription Factors SPATULA, HECATE1, and INDEHISCENT"

_plants, 2025, doi:10.3390/plants14121756_

Round 1

Reviewer 1 Report

Comments and Suggestions for Authors

The manuscript by López-Gómez et al describes the use of AlphaFold and molecular docking structure and interaction predictions for studying protein-protein interactions between three Arabidopsis transcription factors crucial for gynoecium development. On the basis of those predictions, key residues of SPT, HEC1 and IND are mutagenized and the effects studies by BiFC and through overexpression in transgenic plants. The combined use of AphaFold and molecular docking is a relatively new approach that has not been used extensively yet, in particular for plant TFs and in a developmental context. Gynoecium development and TF protein-protein interactions are both very relevant research topics in plant developmental biology. The main point of the manuscript could be that it demonstrates the viability of the computational-based approach for studying plant TF protein-protein interactions, which will be of interest for other researchers on this topic. The functional information gained for SPT, HEC1 and IND is limited in the absence of additional experiments, but this is recognized in the discussion.

The manuscript is well written, clear and concise, and the experiments are all well described. Following are some minor suggestions for consideration:

- 35S::SPTD appears to act as a dominant negative (line 375-378). It would be nice to add spt to Figure 4 if the authors have such data available.

- Could AlphaFold” and molecular docking have been used to predict, model, or evaluate the alterations (in structure, interaction, etc.) that would be caused by the introduced mutations? Would such analysis have suggested a different set of mutations to disrupt the interactions. Perhaps this could be commented on in the discussion.

- Maybe the authors could comment on the possibility of the mutations affecting protein stability?

- Line 193 “there was a notable difference”. It should be “there was not a notable difference”

Author Response

Reviewer 1

Comments and Suggestions for Authors

The manuscript by López-Gómez et al describes the use of AlphaFold and molecular docking structure and interaction predictions for studying protein-protein interactions between three Arabidopsis transcription factors crucial for gynoecium development. On the basis of those predictions, key residues of SPT, HEC1 and IND are mutagenized and the effects studies by BiFC and through overexpression in transgenic plants. The combined use of AphaFold and molecular docking is a relatively new approach that has not been used extensively yet, in particular for plant TFs and in a developmental context. Gynoecium development and TF protein-protein interactions are both very relevant research topics in plant developmental biology. The main point of the manuscript could be that it demonstrates the viability of the computational-based approach for studying plant TF protein-protein interactions, which will be of interest for other researchers on this topic. The functional information gained for SPT, HEC1 and IND is limited in the absence of additional experiments, but this is recognized in the discussion.

The manuscript is well written, clear and concise, and the experiments are all well described. Following are some minor suggestions for consideration:

Response 1: Thank you for your time and kind words.

- 35S::SPTD appears to act as a dominant negative (line 375-378). It would be nice to add spt to Figure 4 if the authors have such data available.

Response 2: Thank you for this comment. We have added images of spt mutant gynoecia in the new Figure S7, comparing it to the wild-type and the mutated SPT version.

- Could AlphaFold” and molecular docking have been used to predict, model, or evaluate the alterations (in structure, interaction, etc.) that would be caused by the introduced mutations? Would such analysis have suggested a different set of mutations to disrupt the interactions. Perhaps this could be commented on in the discussion.

Response 3: This is interesting. In part yes, the mutated sequences can be put back in Alphafold2 for modeling to see if the structure changes. We did this and the results are presented in the new Figure S13. For SPT we noticed changes, but for HEC1 and IND not really. The difference is that for HEC1 and IND the mutations are in the conserved domain. This result is added to the Discussion: page 14, lines 410-413.

We also mentioned in the Results why we choose for Analine substitutions (page 4, lines 142-144), because in principle this does not affect the structure of the protein, so in principle we did not expect many changes. In the end, it is difficult to answer if we made the right choice for the mutations based on looking to the model of the modified sequences; it stays a bit just doing the mutations. Because perhaps it seems that other amino acids are better to mutate because we added various analines to the sequence. Furthermore, we also tried to perform the molecular docking, which can take up to a week or so. However, every time that we tried we obtained an error and the support team was not able to solve the problem. So, unfortunately, we do not have the binding free energy for the amino acids, though it is to expect that when the amino acids change, the free energy will also change. In the time we received for the revision, we will not be able to do this. On the other hand, we added a line in the Discussion on the release of Alphafold3, which allows to perform modeling and docking with the same tool. So, the advancements in this field go very fast. What we can conclude from our work is that the use of these tools allows to identify amino acids that are involved in some way in protein interactions (or other factors such as protein stability suggested by your next question).

- Maybe the authors could comment on the possibility of the mutations affecting protein stability?

Response 4: Good point, though difficult to address in this moment. With the mutated versions, for HEC1 we do not know, because the interaction SPT-HEC1 is lost. SPT-IND is reduced, so at least there should be SPT and IND protein present. Future studies could address this by using antibodies for instance or GFP fusions…. We added a line in the Discussion on this, page 14, line 406.

- Line 193 “there was a notable difference”. It should be “there was not a notable difference”

Response 5: Thanks for catching this error. This has been corrected.

Reviewer 2 Report

Comments and Suggestions for Authors

This work is devoted to the study of protein-protein interactions of transcription factors involved in flower development in Arabidopsis. At the moment the technique of computer prediction of protein structure is developing very intensively. And in the last few years, impressive progress has been made in this area. The authors of this work used the appeared opportunities to establish the amino acids involved directly in the interaction of transcription factors. In general, the work was performed at a good methodological level. Almost a full range of studies was carried out, from computational predictions to testing mutant proteins under constant expression in Arabidopsis plants. Nevertheless, I am left with some questions about this work. The paper does not sufficiently describe the interactions between specific amino acids of the partner proteins. The choice of specifically these amino acids for mutagenesis is not clearly explained. It seems to me that it is not quite correct to use the Δ symbol to denote point substitutions. It still denotes a deletion of a sequence fragment. In the BiFC analysis, no data on protein expression are presented in the paper. Thus, the absence and weakening of the signal can also be explained by a decrease in the level of gene expression. The interaction was investigated by only one method. At the same time, such studies require confirmation by other methods. A rather unexpected result was obtained when these mutant proteins were expressed in Arabidopsis plants. It is almost opposite to the expected one. And the authors put forward an assumption about the obtained effects. But I would like to understand why authors of this work did not try to create compensatory transgenic plants using insertion mutants for the studied genes as genetic background, the existence of which is mentioned at the manuscript, using native promoters? Constitutive expression itself could have been the cause of the observed unexpected effects. Thus, this work needs substantial revision, unless, of course, the authors provide convincing answers to the questions that have arisen.

Author Response

Reviewer 2

Open Review

Comments and Suggestions for Authors

This work is devoted to the study of protein-protein interactions of transcription factors involved in flower development in Arabidopsis. At the moment the technique of computer prediction of protein structure is developing very intensively. And in the last few years, impressive progress has been made in this area. The authors of this work used the appeared opportunities to establish the amino acids involved directly in the interaction of transcription factors. In general, the work was performed at a good methodological level. Almost a full range of studies was carried out, from computational predictions to testing mutant proteins under constant expression in Arabidopsis plants.

Response 1: Thank you for your time and kind words.

Nevertheless, I am left with some questions about this work. The paper does not sufficiently describe the interactions between specific amino acids of the partner proteins. The choice of specifically these amino acids for mutagenesis is not clearly explained.

Response 2: Thank you for this comment. We have added an explanation why we choose for the amino acids to substitute with analine; page 4, lines 136-137. Table 1 and 2 show the most likely amino acids involved in protein-protein interactions. We choose to substitute amino acids with the lowest free binding energy and that where next to each other.

 It seems to me that it is not quite correct to use the Δ symbol to denote point substitutions. It still denotes a deletion of a sequence fragment.

Response 3: Indeed, we agree. We have changed the naming of the versions in the whole manuscript and all the figures. The description of the naming is at page 4, line 140-141:

´ We designated the following names for the mutated versions: SPT-4A, HEC1-2A, and IND-3A´.  The classical way of writing with numbers of the amino acids changed would make very long names.

In the BiFC analysis, no data on protein expression are presented in the paper. Thus, the absence and weakening of the signal can also be explained by a decrease in the level of gene expression.

Response 4: We would need antibodies to verify protein levels, which we do not have. Reduced level of proteins would still show a signal. With the mutated versions, for HEC1 we do not know, because the interaction SPT-HEC1 is lost. SPT-IND is reduced, so at least there should be SPT and IND protein present. Future studies could address this by using antibodies for instance or GFP fusions…. We added a line in the Discussion on this, page 14, line 406. Related to the mentioned gene expression, based on the Arabidopsis experiments, the mutated versions are well expressed in plants.

The interaction was investigated by only one method. At the same time, such studies require confirmation by other methods.

Response 5: This is true, but we used a well-established system in plants, with many repetitions. In the time given to make the revisions (10 days), we are not able to verify this with another system. We recognize the limitation mentioned.

A rather unexpected result was obtained when these mutant proteins were expressed in Arabidopsis plants. It is almost opposite to the expected one. And the authors put forward an assumption about the obtained effects. But I would like to understand why authors of this work did not try to create compensatory transgenic plants using insertion mutants for the studied genes as genetic background, the existence of which is mentioned at the manuscript, using native promoters? Constitutive expression itself could have been the cause of the observed unexpected effects. Thus, this work needs substantial revision, unless, of course, the authors provide convincing answers to the questions that have arisen.

Response 6: Thank you for these observations. It would be interesting to transform the mutated versions to mutants and/or express the versions under their native promotors, as discussed in the Discussion (page 14). To do this would require substantial extra time, and for instance for HEC1, the best would be to use the triple hec1,2,3, mutant due to redundancy, so not so easy. We recognize the comments made and future experiments in that direction will be done, but for this first story, it shows that this approach is promising to find important amino acids.  

Reviewer 3 Report

Comments and Suggestions for Authors

Authors used bioinformatic tools to model protein structures of three the bHLH trans-factors SPATULA (SPT), HECATE1 (HEC1), and INDEHISCENT. Obtained structures were used to protein-protein docking and description of aa residues of critical importance for the interaction using the alanine scan.

Protein –protein interactions were tested in plants transformed by protein mutants. The impact of mutants on plant gynoecium and fruit development was stududied in plants overexpressing the mutated proteins.

The study provides information related to protein-protein interaction that are critical for gynoecium and fruit development. Results are novel and could be interesting to researchers in the field. Study is well planned and performed, obtained results support conclusions. Figures are of good quality.

The relatively weaker part is the lack of independent methods, for example PCR of genomic DNA, used for plant transformation confirmation. Also the protein models are not of high quality.

Following comments should be addressed to further  improve the manuscript:

Tables 1 and 2.

If possible to perform using available software, try to show the impact of substitution by Ala on the binding free energy – the less negative or positive value could be expected.

Secton 4.3.

Add information of the difference between VN-GW and VC-GW vectors. How the use of different vectors could affect the observed protein-protein interactions.

Section 4.5

Provide details of PCR cycling reaction.

Add name, manufacturer and country of origin of used RT-PCR equipment.

Provide approximate amount of RNA per one qPCR sample.

Table S2- add the length of PCR amplicons. Add the sequence of reference gene primer pair and the length of amplicon.

Add information of software used to acquisition of raw RT-PCR data.

Author Response

Reviewer 3

Comments and Suggestions for Authors

Authors used bioinformatic tools to model protein structures of three the bHLH trans-factors SPATULA (SPT), HECATE1 (HEC1), and INDEHISCENT. Obtained structures were used to protein-protein docking and description of aa residues of critical importance for the interaction using the alanine scan.

Protein –protein interactions were tested in plants transformed by protein mutants. The impact of mutants on plant gynoecium and fruit development was stududied in plants overexpressing the mutated proteins.

The study provides information related to protein-protein interaction that are critical for gynoecium and fruit development. Results are novel and could be interesting to researchers in the field. Study is well planned and performed, obtained results support conclusions. Figures are of good quality.

The relatively weaker part is the lack of independent methods, for example PCR of genomic DNA, used for plant transformation confirmation. Also the protein models are not of high quality.

Response 1: Thank you for your time and kind words.

Following comments should be addressed to further  improve the manuscript:

Tables 1 and 2.

If possible to perform using available software, try to show the impact of substitution by Ala on the binding free energy – the less negative or positive value could be expected.

Response 2: Thank you for this comment. Indeed, with other amino acids, the binding free energy will change. This question is related to Reviewer 1, to do modeling again with the mutated versions , as well as the docking. Our answers are: the mutated sequences can be put back in Alphafold2 for modeling to see if the structure changes. We did this and the results are presented in the new Figure S13. For SPT we noticed changes, but for HEC1 and IND not really. The difference is that for HEC1 and IND the mutations are in the conserved domain. This result is added to the Discussion: page 14, lines 410-413.

We also mentioned in the Results why we choose for Analine substitutions (page 4, lines 142-144), because in principle this does not affect the structure of the protein, so in principle we did not expect many changes. In the end, it is difficult to answer if we made the right choice for the mutations based on looking to the model of the modified sequences; it stays a bit just doing the mutations. Because perhaps it seems that other amino acids are better to mutate because we added various analines to the sequence. Furthermore, we also tried to perform the molecular docking, which can take up to a week or so. However, every time that we tried we obtained an error and the support team was not able to solve the problem. So, unfortunately, we do not have the binding free energy for the amino acids, though it is to expect that when the amino acids change, the free energy will also change. In the time we received for the revision, we will not be able to do this. On the other hand, we added a line in the Discussion on the release of Alphafold3, which allows to perform modeling and docking with the same tool. So, the advancements in this field go very fast. What we can conclude from our work is that the use of these tools allows to identify amino acids that are involved in some way in protein interactions.

Secton 4.3.

Add information of the difference between VN-GW and VC-GW vectors. How the use of different vectors could affect the observed protein-protein interactions.

Response 3: Thank you. We have added a description of the vectors; page 16, lines 469-473:

´ …….clones were recombined with the Gateway-compatible destination vectors pVN/gw and pVC/gw [60]. These vectors contain the N-terminal fragment comprised of residues 1 to 154 (YN) and the C-terminal fragment comprised of residues 155 to 238 (YC) of the Venus YFP variant, respectively. The backbone of these vectors is the transient expression pBI221 plasmid with the 35S promoter [60]´.

Section 4.5

Provide details of PCR cycling reaction.

Response 4: 25x cycles, added to the text.

Add name, manufacturer and country of origin of used RT-PCR equipment.

Response 4: Done. Thermocycler MiniAmp™ Plus (Applied Biosystems).

Provide approximate amount of RNA per one qPCR sample.

Response 5: Mentioned in M&M, around 1 ug of total RNA was used to make cDNA. Page 17, line 513.

Table S2- add the length of PCR amplicons. Add the sequence of reference gene primer pair and the length of amplicon.

Response 6: Thank you. We added this to Table S2.

Add information of software used to acquisition of raw RT-PCR data.

Response 7: Done. ImageJ (https://imagej.net/ij/) was used to PCR band intensities.

Round 2

Reviewer 2 Report

Comments and Suggestions for Authors

The answers of the authors satisfy me.

Reviewer 3 Report

Comments and Suggestions for Authors

Authors answered to all  questions and corrected teh manuscript as required. I have no other comments.